



# Towards understanding the influence of seasons on low groundwater periods based on explainable machine learning

Andreas Wunsch[1,2], Tanja Liesch[2], Nico Goldscheider[2]

[1]Fraunhofer Institute of Optronics, System Technologies and Image Exploitation IOSB, Karlsruhe, Germany
[2]Karlsruhe Institute of Technology, Karlsruhe, Germany

*Correspondence to*: Andreas Wunsch (andreas.wunsch.edu@gmail.com)

**Abstract.** Seasons are known to have a major influence on groundwater recharge and therefore groundwater levels, however, underlying relationships are complex and partly unknown. The goal of this study is to investigate the influence of the seasons on groundwater levels (GWL), especially on low-water periods. For this purpose, we train artificial neural networks and apply

layer-wise relevance propagation to understand what relationships are learned by the models to simulate GWLs. We find that the learned relationships are plausible and thus consistent with our understanding of the major physical processes. Our results show that the models learn summer as the key season for periods of low GWL in fall, a connection to the preceding winter is usually only subordinate. Specifically, dry summers show strong influence on low-water periods and generate a water deficit, that (preceding) wet winters cannot compensate. Temperature is, thus an important proxy for evapotranspiration in summer

and overall identified as the more important variable, but only on average. Single precipitation events show by far the highest influences on GWL and summer precipitation seems to mainly control the severeness of low GWL periods in fall, while higher summer temperatures do not systematically cause more severe low-water periods.

## 1 Introduction

Groundwater is a major source of drinking water globally, and is also used for agricultural irrigation, industrial purpose and to

supply terrestrial and aquatic groundwater-dependent ecosystems (Gleeson et al., 2016; Siebert et al., 2010). However, groundwater resources are under increasing pressure resulting from climate change, intensified land use and increasing groundwater abstraction (Famiglietti, 2014; Green et al., 2011). Low-water periods are thereby of particular interest since they often cause problems, e.g., for groundwater dependent ecosystems or water supply. Moreover, they mostly coincide in time with periods of higher water demand and therefore increased abstraction rates, which exacerbates the problem. The sustainable

availability of groundwater resources is chiefly determined by groundwater recharge. Overexploitation occurs when abstraction exceeds recharge. Recharge is difficult to quantify directly and precisely on large areas, but in shallow, unconfined, and unused aquifers, groundwater levels (GWL) are a good proxy for recharge, and changes in groundwater level are a straightforward way to quantify changes in groundwater availability (Hartmann et al., 2012). On long time scales, recharge is the difference between precipitation and actual evapotranspiration (minus overland flow if present); on shorter time scales,





changes in soil moisture storage play a major role in recharge and, consequently, groundwater levels. During the vegetation period, most of the precipitation is used by the vegetation for evapotranspiration. After long dry periods, large quantities or rainfall are needed to replenish the soil water deficit, before recharge can start (Döll and Fiedler, 2008). In the cold season, however, when soils are typically water saturated, most of the precipitation water is available for recharge, unless it is stored in the snow cover (Petitta et al., 2022).

These generalized relations show that the seasons have a major impact on groundwater recharge, although the underlying processes and relationships are quite complex and still not completely understood. From glaciology, it is known that the summer season often has a larger impact on glacier retreat that the winter season (Fujita and Ageta, 2000; Thibert et al., 2013; Trachsel and Nesje, 2015). To put it simply, a long, hot, and dry summer can cause more damage to the glacier than a long winter with plenty of snow can repair. Similar relationships have been observed in soil science, where long-term lysimeter

data have shown that hot, dry summers have a much greater negative impact on soil water storage than the positive influence of a wet winter season (Merk et al., 2021). The principal goal of this study is to investigate the influence of the seasons on groundwater levels, especially on low water periods, and our initial hypothesis is that hot dry summers have a stronger negative influence on groundwater resources than could be compensated for by (preceding) wet winters.

Data-driven groundwater modelling based on Machine Learning (ML) methods is now an established yet still emerging field,

as shown in a recent review by Tao et al. (2022). The ability of ML models to simulate GWLs based on historic groundwater and meteorological data alone and without comprehensive knowledge and data of the underground structure makes them appealing methods compared to physically based and numerical methods (Adamowski and Chan, 2011), and it was found that Artificial Intelligence (AI) methods (including ML) can successfully be used to simulate and predict GWL time series in different aquifers (Rajaee et al., 2019). Despite their success in terms of good model performance, one often mentioned

drawback of AI/ML models is their "black-box" characteristics, as they do not rely on known physical relationships. However, explainable AI (XAI) methods can help to overcome this problem. They allow to interpret model behavior, and thus not only build trust in the models, but also may help to get new insights that are not apparent from the data alone. A good overview of XAI methods, including their history, motivation, goals, and types is given by Samek et al. (2019) and Holzinger et al. (2022). Popular types range from surrogate functions (e.g., Local Interpretable Model-agnostic Explanations (LIME) (Ribeiro et al.

2016)), local perturbation based (sensitivity) methods (e.g., SHapley Additive exPlanations (SHAP) (Lundberg and Lee, 2017)) to propagation-based approaches, which integrate the internal structure of the model into the explanation process. Layer-wise Relevance Propagation (LRP) (Bach et al., 2015; Montavon et al., 2019) is a propagation-based explanation framework, which is applicable to artificial neural networks (ANN). It decomposes the output of the nonlinear decision function in terms of the input variables, forming a vector of input features scores that constitute the 'explanation' (Lapuschkin

et al., 2019). LRP has been extensively applied and validated in numerous disciplines including computer vision, medicine, natural language processing, economy, and others. However, to the best of our knowledge the application in earth science is limited to Toms et al. (2020) and Mirzavand Borujeni et al. (2023), who use it in the context of El Niño–Southern Oscillation and surface sea temperature forecasts, as wells as air pollution, respectively. We chose this method, since it is rather





straightforward, easy to understand and interpret, and applicable to sequence-alike/time series input data with deep learning
models. Moreover, it has some advantages to other XAI methods such as its high computational efficiency and its theoretical
underpinning based on Deep Taylor Composition (Montavon et al., 2017), making it a trustworthy and robust explanation
method (Arras et al., 2022).

This study aims to explore different research questions:

1.  Is it possible to use LRP to explore what ANNs learn when simulating GWLs with meteorological input data, and
70          disentangle the temporal component of such learned relationships?

2.  Do these relationships coincide with our existing conceptual understanding of the relevant processes?

3.  What do the models identify as key drivers for periods of low GWL?

4.  What is the specific influence of each season and the temporal precipitation and temperature patterns during these
seasons?

To answer these questions, we train one-dimensional convolutional neural networks (CNN) at 24 example locations spread
throughout Germany and apply LRP to explore what these models learn when they receive meteorological input data to
simulate groundwater levels over time. In terms of model choice, we prefer CNNs over recurrent alternatives such as long
short-term memory networks (LSTM) (Hochreiter and Schmidhuber, 1997), because they proved to be well suited and reliable
in earlier studies (e.g., Wunsch et al., 2022). As input forcing data, we use exclusively precipitation and temperature, which
yields good simulation results, and, due to the low variable number, simplifies later interpretation of the learned relationships.

## 2 Data and Methods

### 2.1 Data and locations

In this study we use groundwater data from 24 different locations throughout Germany. All locations represent the uppermost,
unconfined aquifer and exhibit weekly groundwater timeseries with a minimum length of 24 years (1997-2020) up to 66 years
(1955-2020). Most wells are located in very shallow porous aquifers; two wells each are, however, located in fractured and
karst aquifers, with slightly higher depth to groundwater. The locations, the start year of the weekly data records, the aquifer
type, and the depth to groundwater are depicted in Fig. 1. The groundwater data until 2015 are a subset of publicly available
data (Wunsch et al., 2021b) and preprocessed as described in Wunsch et al. (2022). More recent data were added using openly
available and gapless groundwater data from the respective online services of the federal environmental agencies.

Input data are precipitation and temperature from the respective locations within the HYRAS v5.0 dataset by the German
Meteorological Service (Rauthe et al., 2013; Razafimaharo et al., 2020). The HYRAS v5.0 dataset is a downscaled raster
dataset with a cell size of 1 km², based on observations from meteorological stations, and is openly available via DWD
Opendata (2022). Conceptually, precipitation serves as a proxy for potential groundwater recharge after compensating deficits
of soil water, while temperature represents evapotranspiration processes. Usually, higher temperature also means higher
evapotranspiration and thus less potential groundwater recharge, however, the relationships are complex and partly dependent.





For example, in winter, higher temperature often goes along with higher precipitation intensity, thus higher potential recharge, because very cold conditions (<< 0°C) are usually dry, whereas in summer precipitation intensity decreases with increasing temperatures (e.g., Berg et al., 2009).

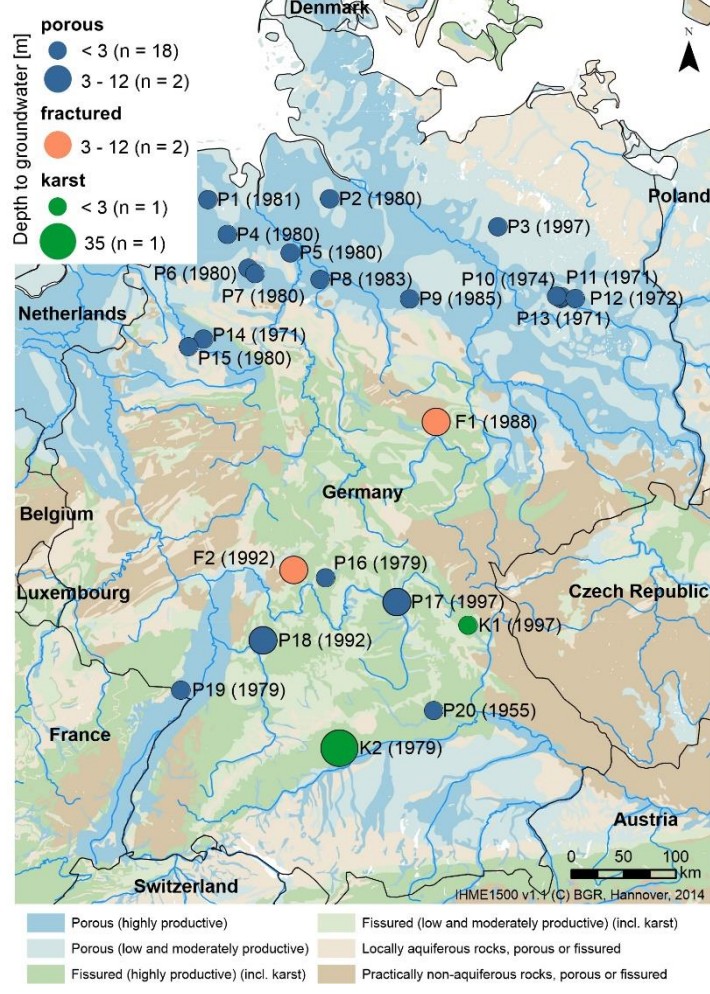

**Figure 1:** Map of the considered locations in Germany, depicting the aquifer type of each well (color), the depth to groundwater (symbol size) and an ID with the start year of the data records in parenthesis (label). The background shows the aquifer type based on IMHE.

## 2.2 Model selection and evaluation

To perform this study, we use convolutional neural networks (CNN) (LeCun et al., 2015), which are commonly applied to image-like data but have also shown to be valuable for the simulation of sequential data such as water related time series (Duan et al., 2020; Wunsch et al., 2021a, 2022). The CNNs applied in this study comprise the layers shown in Fig. 2 and use the hyperparameters listed in Table 1. All models are applied in a sequence-to-value forecasting mode and use a fixed input sequence length of 52 weeks (1 year), as illustrated in Fig. 2. This is necessary to answer the research questions of this study, and to enable comparability between models. A Bayesian optimization (Nogueira, 2014) is applied to select the optimal





configuration for training batch size, number of filters in the 1D convolutional layer and the number of neurons in the first
dense layer according to the range listed in Table 1. Between 80 and 200 optimization steps are performed, above 80 the
process stops if no improvement occurs for 25 steps. Because the models depend on a random initialization, we use a model
ensemble of 20 independently trained CNNs (only 5 for each optimization step to save computation time). We derive a 90%
confidence interval from the model ensemble based on these 20 model initializations, meaning that 18 of 20 model runs lie
within the shown interval. All models are implemented in Python 3.8, using TensorFlow 2.7 (Abadi et al., 2015), Keras
(Chollet, 2015) and the libraries Numpy (van der Walt et al., 2011), Pandas (Reback et al., 2020), Scikit-learn (Pedregosa et
al., 2011), and Matplotlib (Hunter, 2007).

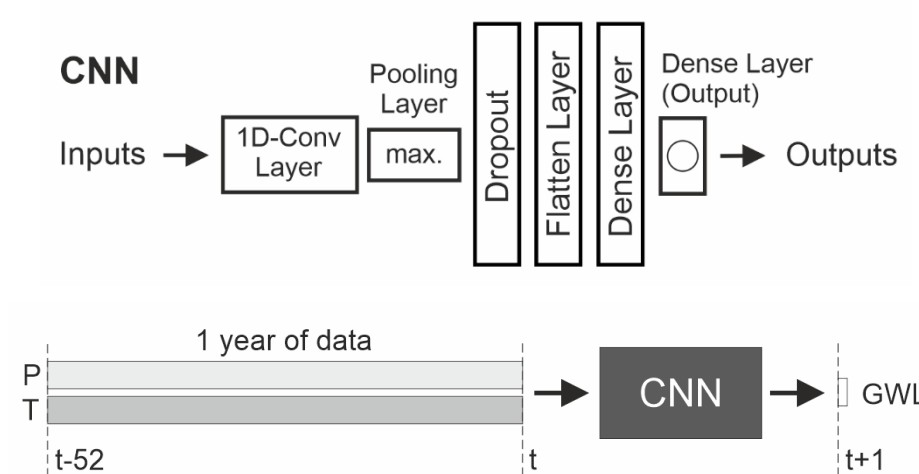

**Figure 2:** Structure of the CNN models (upper part) and illustration of the sequence-to-value forecasting mode with a fixed input sequence
of one year (lower part).

**Table 1:** Summary of model hyperparameters and important parts of the modelling and evaluation strategy.

| Hyperparameters (fixed) | |
| --- | --- |
| Length of input sequence | 52 steps (1 year) |
| 1D-Convolution kernel size | 3 |
| Dropout rate | 10 % |
| Loss function | Mean squared error (MSE) |
| Optimizer (initial learning rate) | ADAM (0.001) |
| Max. training epochs | 500 |
| Early stopping patience | 30 |
| **Hyperparameters (optimized)** | **Range** |





| Batch size | between $2^4$ (16) and $2^9$ (512) |
|---|---|
| Size of first dense layer | between $2^4$ (16) and $2^8$ (256) |
| Number of 1D-convolution filters | between $2^4$ (16) and $2^9$ (512) |
| **Training and optimization strategy** | |
| Optimization period | 2015 – 2016 |
| Testing period | 2017 – 2020 |
| Training and early stopping (splitting ratio) | Before 2015 (90% / 10%) |
| Bayesian optimization steps (Min, Max) | 80, 200 |
| Size of model ensemble (pseudorandom): | |
| during optimization | 5 |
| final | 20 |
| Optimization target | MSE |

We selected only those locations where the tested models achieve particularly good scores in the test set (Fig. 3b, details on
each location in the supplementary material). This way, we reduce uncertainty from model inaccuracies during the following
analyses. However, because we will analyze the model not only in the test period, but selected periods of the complete
individual time series, we explored the model fit for the full time series and selected only locations with a highly accurate fit
throughout the complete simulation (Fig. 3a). The simulation accuracy is demonstrated in Fig. 3 using Nash-Sutcliffe
Efficiency (NSE) (Nash and Sutcliffe, 1970), coefficient of determination (R²) and Kling-Gupta Efficiency (Gupta et al., 2009).
We further rigorously judged the fit between observed and simulated values in all parts visually, to reduce possible influence
of counterbalancing error effects. An example simulation and an illustration of the time series partition for training,
optimization and testing is depicted in Fig. 4 for location P11.

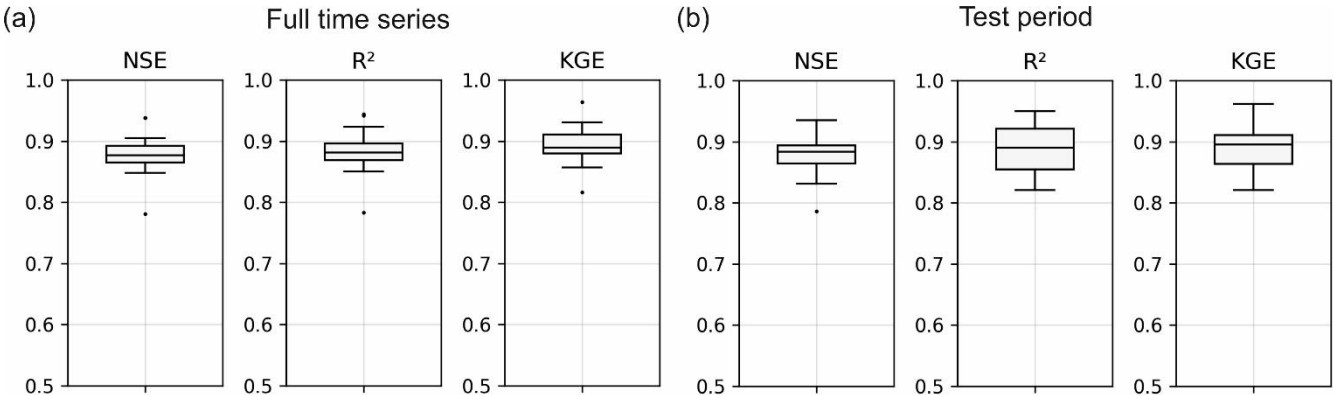

**Figure 3:** Model performance for the complete time series (a) and for the test period only (b) (at all 24 locations).





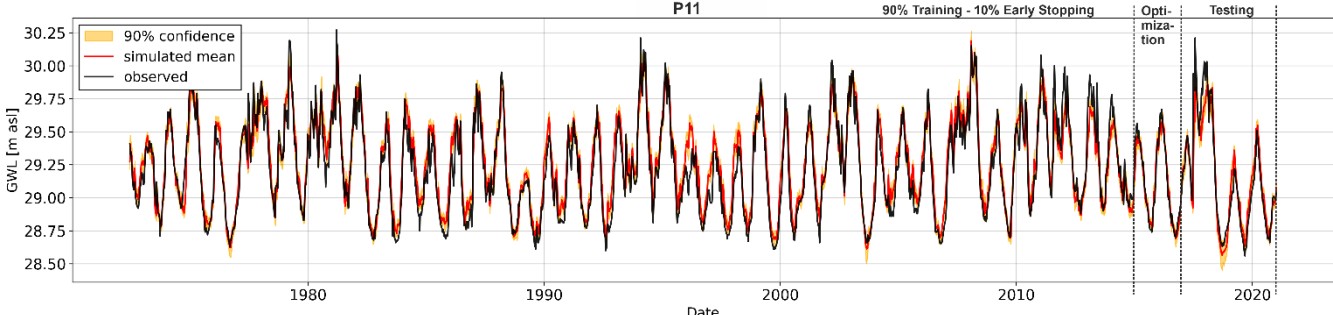

**Figure 4:** Model fit with high accuracy for all parts of the respective time series (location: P11, compare Fig. 1).

## 2.3 Layer-Wise Relevance Propagation

Layer-wise relevance propagation (LRP) (Bach et al., 2015) is a framework to explain model predictions by decomposition. LRP redistributes the prediction $f(x)$ backwards throughout all layers of a neural network (in our case) using local redistribution

rules and assigns a relevance score $R_i$ to each input (Samek et al., 2017), hence in in our case a score for each value within the input sequence of both input variables P and T is calculated. LRP further is a local explanation method that explains each prediction using a single set of inputs. An important part of LRP is the conservation property, which means that each $R_i$ of each input determines its individual contribution to the model output $f(x)$, and no relevance is added or removed during the relevance redistribution procedure (Samek et al., 2017). LRP thus exhibits the additive feature attribution property, which

means the sum of all $R_i(x)$ equals $f(x)$. Several redistribution or attribution rules exist, the most basic one is the LRP$_z$-rule, which performs a proportional decomposition and which we use in this study (e.g., Kohlbrenner et al., 2020). We implement LRP using the *iNNvestigate* toolbox by Alber et al. (2019). As we use a model ensemble of 20 CNNs per location, for each individual $R_i$ value we investigate the mean $R_i$ value of all 20 models for further interpretation during our analyses.

## 3 Results and Discussion

For the following analyses, we refer to the four seasons as the 3-month periods D-J-F (winter), M-A-M (spring), J-J-A (summer) and S-O-N (fall). At the investigated locations the annual minimum usually occurs during September, which is why we distinguish between summer (JJA) and a so-called low-water period that we define as the 3 months from July to September (JAS). Besides the annual minimum, this period also nicely catches the strongest downward trends of the considered groundwater hydrographs. The corresponding high-water period that includes the annual maximum in January or February,

and the strongest increasing groundwater levels of the annual cycle equals the winter period and does not need a separate definition.

In the following we explore the influence of the four seasons on those low-water periods. Thanks to the additive feature attribution property of LRP, we can sum all $R_i$ within a certain time period (here one season) in the input sequence of a simulated groundwater level in a low-water period, to estimate the effect of the whole season on the model output. The results





for all low-water periods at all locations are shown in Fig. 5. For spring, summer, and fall we mostly find negative contributions of T (i.e., higher temperatures relate to lower GWL) and positive contributions of P (i.e., higher precipitation coincide with higher GWL), as it can be expected. We see that summer (Fig. 5c) has the largest (generally high absolute relevance scores $R_i$) and winter (Fig. 5a) has the smallest influence on the GWLs in low-water periods, while spring and fall contributions are moderate. In winter (in parts also in spring and fall), T predominantly contributes slightly positive, while negative contributions

are subordinate. This might be explained by correlation effects of T and P, e.g., higher temperatures in winter and some periods of spring and fall are often associated with higher rainfall, and, especially in winter, low temperatures can be associated with either snow (which is included in P, but does not directly lead to a groundwater level increase due to snow storage) or rather dry periods (Berg et al., 2009; Trenberth and Shea, 2005). The influence of summer is plausible, both, in its relative strength because of the temporal proximity (overlap even), as well as its clear positive contributions of P and negative contributions of

T. However, the small contribution values in winter demonstrate that the models do not learn any strong connection between winter and low-water period, which also means that a preceding wet winter does not seem to be able to compensate for the negative influence of a following summer. Both spring and fall show similarly moderate influence. The influence of fall is even higher than of winter, despite the longer time lag and might be related to the model learning that the conditions one year earlier have a certain importance. However, our approach per se cannot account for accumulating effects over several years,

which is a clear limitation. Especially in summer, the influence of P can be clearly distinguished between high groundwater levels (blue dots) and low groundwater levels (red dots, i.e., a spread of red and blue points along the P-axis), while the influence of T is rather uniform. That leads to the conclusion that the models learn summer P as the control for the severeness of a low-water period, whereas the temperature has a generally strong negative influence, but it cannot be seen that higher T lead to predominantly lower groundwater levels in dry periods.

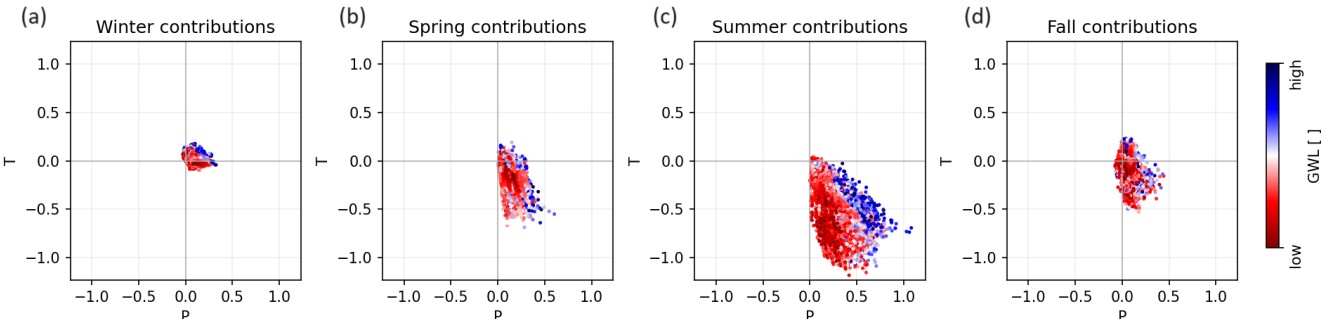

**Figure 5:** Influence of seasons on low-water periods, distinguished by input variable.

In the following, we take a closer look on the generally identified influence of the input variables on groundwater levels (Fig. 6). In contrast to the analysis above, single events (data points) are shown, not sums within specific periods. The x-axis represents the contribution to the model output, the color encodes the input feature value. We find results in agreement with

the analyses above, meaning that LRP identifies T as on average more important than P (thus T is on top of the plot), T is clearly responsible for negative contributions, and P contributes mostly positively to the model output. P exhibits a clear





positive correlation with the relevance scores (Pearson r = 0.60, p = 0.0), meaning that strong P events contribute stronger positively to the model output than weak events. The negative influence of T is less clear in this sense, and we find only a weak negative correlation (Pearson r = -0.14, p = 0.0). The reason for this could be the partly contradictory role of temperature

depending on the season, as already discussed in the context of the positive contributions of T in winter in Fig. 5a. In contrast to the analyses shown in Fig. 5, where the maximum importance values are higher for T than for P, we now look at single events, and here we clearly see that, in absolute values, strong precipitation contributes up to twice as strong compared to temperature. Note, that a few LRP relevance scores for high P inputs (dark blue) exhibit negative values. Further investigation showed that these occur predominantly with a large temporal distance to the target. It thus might be a way of the model to cope

with strong precipitation events in the past that do not influence the model output positively anymore. We speculate that this might be an effect of the long input sequence that we forced the model to use and that most certainly is longer than an optimization would have selected for the respective location.

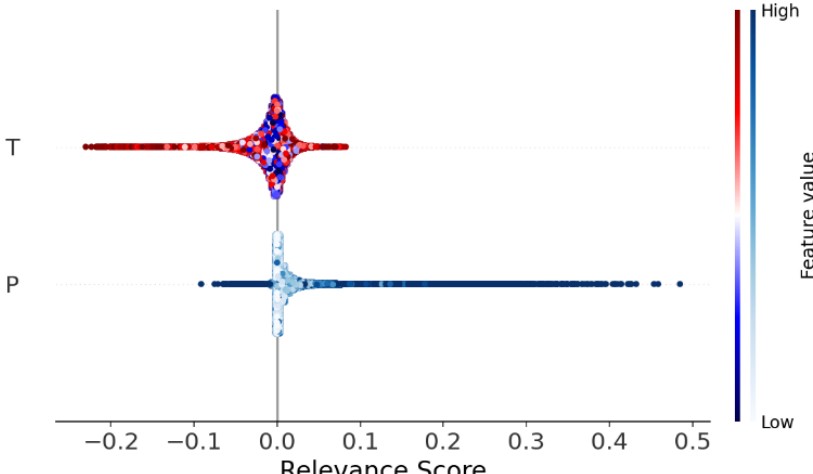

**Figure 6:** Summary bee-swarm plots for all locations to show the learned relationships between input variables (P, T) and groundwater
levels.

In the following we explore the results at location P11 in detail, which in terms of results is a typical example from our dataset. Figure 7 shows the raw data of the former analyses, thus input data and corresponding LRP values, temporally ordered for the test period and should be read as follows: (i) Subplot (c) shows the observed and simulated GWL within the test period. (ii) Each simulated GWL (e.g., at time $t_1$ or $t_2$) is based on input data of one year (52 values). Such raw input data is displayed

above in subplots (a) and (b) for P and T respectively. (iii) Additionally, heatmaps in (a) and (b) show the LRP relevance scores for each input sequence (lines), from t-52 (left edge) to t (right edge). All subplots share the same x-axis and are aligned in time. Corresponding figures for all other locations are part of the supplementary material.

Figure 7 visualizes well how LRP importance changes for each input value over time within the input sequence. For all P events, the heatmap of LRP values shows that blue fades out in columns from top to bottom, meaning the importance of P

events decreases with the temporal distance to the target value (right edge), which is a plausible behavior. Even though some





events (e.g., July 2017) do not seem to decrease, in reality they do, and it is only an effect of the upper limit of the color scale. Overall, strong events have an influence that lasts longer than that of weak events. We find that all LRP relevance scores in (a) are either positive or close to zero, while negative influences (as in Fig. 6) are not visible for P11. The above-described seasonal differences in P contributions are also not clearly visible for P11.

The second heatmap of LRP values of T inputs in (b), shows that summer T causes stronger negative contributions than winter T in the close past, causing a periodical color pattern of dark and light red on the right edge of the diagonal. In contrast, with larger temporal distance to the target values (middle part of diagonal), all T inputs cause neutral (white) or even slightly positive (blue) contributions. Again, summer T causing stronger LRP contributions compared to winter T, however more positive in this case (white/blue diagonal). While this is only one example location, we find such patterns (summer winter periodicity
and/or negative contribution changing to positive contribution with temporal distance) regularly in our data. When investigating a particularly low groundwater level in late September 2018 ($t_1$) and examining the LRP values of the relevant input sequence, we find that though the temperatures were on average higher in the months before, the T-LRP values show only moderately negative influences in this time period. Rather the temperature in the winter before has slightly lower positive influence, and there were exceptionally few precipitation events in the relevant time period. When looking at the low-water
period one year before ($t_2$), which exhibits a distinctly higher groundwater level than $t_1$, the LRP values of T in the weeks before are much more negative, but obviously were counteracted by heavy rainfall events in summer 2017, also shown by strong positive LRP values for P. This confirms the results shown above, that summer P seems to be the most dominating factor for low groundwater levels in late-summer low-water periods.








**Figure 7:** Breakdown of the LRP importance values of each variable in the input sequence within the test set (2017-2020) at location P11. All graphs are aligned in time (x-axis). The dashed lines indicate how to read the figure. Each forecasted GWL (c) at an arbitrary point in time (e.g., $t_1$ or $t_2$), uses an input sequence of one year (52 values) (compare raw data plots above in (a) and (b)). Hence each horizontal line within the LRP heatmaps for P (a) and T (b) represents the LRP relevance scores for each input value within one input sequence.

By selecting specific periods and rearranging these LRP data, we can get further insights about the differences between drier and wetter low-water periods. Figure 8 thus shows an analysis of the three wettest and the three driest low-water periods at location P11. On top of the figure, seasonal P-sums (a) and T-means (b) are shown, gray bars mark the six selected periods evaluated below. Subplot (c) displays observed and modeled GWLs, also highlighting the selected periods in red and blue.

P11 is a typical example where the low-water periods are dominated by summer P. We find considerably higher LRP values for P in the close past of the wetter low-water periods (e1), compared to the drier periods where the LRP values remain predominantly low (d1). This observation is in agreement with subplot (a1), where we can find the drier periods to have P





sums during summer below average of all years, whereas the wetter periods indeed show sums of at about average (1972, 1974) or above (1977). Mean summer Ts are mild for the wetter low-water periods, in winter, there is no clear systematic for

both P and T (b). Correspondingly, the drier periods exhibit clearly below average summer P (a), while winter P and T, again, show no systematic behavior (a, b). Interestingly we can find stronger negative contributions in terms of T for the wetter years (d2), however the general shapes of (d2) and (e2) are similar, with strong negative values in the close past (spring/summer), neutral values during winter (approx. weeks between 20 and 40) and slight negative values in the far past. Corresponding figures for all other locations are part of the supplementary material to this study.





**Figure 8:** Seasonal P sum (a) and mean T (b) as well as observed and modelled GWLs (c) for the whole time series, and LRP importance of all input sequences for the three wettest (e) and driest (d) low-water periods respectively, separated by input variable. Location: P11.






## 4 Conclusions

In this study we gained insights into the influence of seasons on groundwater levels in Germany, with an emphasis on low-water periods. Layer-wise relevance propagation (LRP), a powerful XAI method, allowed to interpret what artificial neural network models learn regarding the contribution of the two input variables precipitation and temperature in each season.

We found that LRP is a valuable tool to not only gain general insights in what ANNs learn, but also to disentangle such knowledge in time and thus to analyze time series models. In the specific context of GW simulation, we found that the learned relationships do well coincide with the existing conceptual understanding of the relevant physical processes. This makes such modeling results trustworthy and allows to confidentially interpret also yet unknown effects and relationships that can be found in the results.

We find that summer is the key season for low GWL periods. Especially summer precipitation seems to control the severeness of such low water periods in late summer, whereas higher summer T does not per se lead to lower GWL in fall. Wetter low-water periods result from higher summer precipitation and are only subordinately related to the preceding winter season, because, generally, winter exhibits only a minor influence on low GWL periods in late-summer. In summary, dry summers have a major influence on low-water periods and generate a deficit that apparently preceding wet winters cannot compensate for.

In agreement with other studies (e.g., Thober et al., 2018) that indicate that a lower water availability primarily originates from changes in temperature, in this study; T is identified as on average the more important variable. However, this seems only to be the case on average since single P events show twice as high LRP contributions than T in its maximum. The higher influence of P is especially relevant for low-water periods in late summer.

The main limitation of the approach used in this paper is that it cannot account for accumulating effects over several years, as only one year of input data for each forecast is used and the model does not contain any kind of memory. Future research should focus on such inter-annual relationships and should account for such accumulating effects, which of course also complicates evaluation and interpretation. This could be done, e.g., by using recurrent neural networks, which contain a memory state, or replacing the whole ANN-XAI approach with a model class that has better capabilities in this sense.

## Code availability

All Python code necessary to reproduce the results, readily trained models and files containing the results, are either directly provided on Github (https://github.com/AndreasWunsch/influence-of-seasons-on-low-GW-periods, https://doi.org/10.5281/zenodo.10156637) or referenced therein and openly available on Zenodo (https://doi.org/10.5281/zenodo.10156582).



**Data availability**

The groundwater data until 2015 are a subset of publicly available data (Wunsch et al., 2021b). More recent data were added using openly available and gapless groundwater data from the respective online services of the federal environmental agencies. Specific sources are listed in the code repository.

**Supplement link**

https://doi.org/10.5281/zenodo.10157406

**Author contribution**

All three authors Andreas Wunsch (AW), Tanja Liesch (TL) and Nico Goldscheider (NG) conceptualized the study together. AW performed data curation, formal analysis, and investigation. Both AW and TL developed the methodology. AW wrote the software, validated, and visualized the results. TL and NG supervised the work. All three authors contributed substantially to writing the original draft, as well as reviewing and editing.

**Competing interests**

The authors declare that they have no conflict of interest.

**Acknowledgments**

The authors acknowledge support by the state of Baden-Württemberg through bwHPC.

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
