# Peer review of "Towards understanding the influence of seasons on low groundwater periods based on explainable machine learning"

_Hydrology and Earth System Sciences, 2023_

## Referee Comment (RC2)

**Review of: Towards understanding the influence of seasons on low groundwater periods based on explainable machine learning**

The paper gives interesting insights into seasonal patterns of groundwater levels, with the focus on low groundwater level time periods. I think the paper is well written and I have read it with interest. The authors use a machine learning method (LRP) that helps them to evaluate which impact the two meteorological inputs they use (precipitation and temperature) have on low groundwater periods within Germany. They have basically found that average temperature has a bigger impact on low groundwater levels than average precipitation, but single precipitation events show the strongest impact. They have found memory effects, like strong precipitation during the preceding winter, to be of less importance.

I think the motivation for this work is very clear, as we obviously want to deepen our understanding about low groundwater periods and the relevant inputs and timeframes behind them. The paper is well structured and concise and the figures are carefully designed and meaningful. The choice which figures are included in the paper and which are shared as supplementary material was made well and the supplementary material is useful and clearly structured. The title is appropriate and the language is good. The LRP method is presented in a plausible way, but I cannot give an in-depth evaluation on that, since I do not come from a machine learning research background. I think that the paper is a good contribution to HESS and recommend that it will be published after minor revisions, which I will elaborate on in the following. I will start with some general comments which include my most essential remarks and then I will get into detailed comments where I reference specific lines.

General comments:

One key conclusion of this work is that temperature is on average more important than precipitation (stated in the abstract and lines 185 and 269). I do not fully understand where this conclusion comes from. I assume one could get to that conclusion from Figure 5, because values are more distributed along the y-axis than the x-axis? I do not think this is clearly stated within the paper (at least not in a way that reached me) and the sentence in line 185 "LRP identifies T as on average more important than P (thus T is on top of the plot)" confuses me. How does it identify T as more important on average? Could this maybe be expressed / quantified by a single value that is then given to the reader?

I think that the general claim that dry summers have a large impact on low-water periods and wet winters contribute less to these periods is supported by their findings. However, I think this should be relativized, since I would expect that the amount of impact of the preceding winter depends a lot on the catchment and specific location that is observed. Properties of the soil (including the vadose zone) and delayed release of water by other storages like snow could matter much more at different locations and for different water table depths (majority of the observation wells are at very shallow locations).

The term "importance value" is used sometimes within this work, but it is not explained. If it is the same as relevance score, I suggest sticking to one term. If not, I think it needs to be explained.

Detailed comments:

Abstract: I think it should be stated that you use observations wells distributed over Germany and that they are mostly located in very shallow aquifers.

Lines 9-10: I think it should be stated that you use precipitation and temperature as input data.

Line 15: Writing "more important than precipitation" instead of "the more important variable" would improve understandability.

Lines 26-28: I disagree with this sentence. Groundwater levels play an important role when estimating recharge, but they do not represent recharge.

Lines 28-29: I suggest "recharge can be approximated by" instead of "recharge is".

Lines 29-30: I suggest "changes in soil moisture and groundwater storage".

Figure 1 - legend: Without the information in the brackets (n = 18) etc. and with the unit directly behind the number (e.g. < 3 m) I would have understood this legend quicker.

Figure 3 – caption: I suggest removing brackets around "at all 24 locations".

Lines 165- 168: I agree and would like to add that higher temperatures in winter can also cause the snow storage to melt, which might also play a role with respect to the positive contribution of T.

Lines 172-174: If the model learns that conditions one year earlier are important, why does it learn that for fall, but not for winter?

Lines 177-179: I needed to reread this sentence many times, because of the second part of it. I think I understood it now and would recommend writing "it cannot be seen that higher T during dry periods lead to predominantly lower groundwater levels". The way it was phrased before it did not make sense to me, because you always refer to water levels during the low-water period (JAS) in Figure 5 (at least that's how I understood it).

Figure 5: I think it should be stated what we are looking at here (relevance score).

Figures 5 and 6: I think it should be clarified what exactly one dot represents (e.g. one dot per week and location). I think there should be actual values at the colorbar.

Line 211: Judging from this sentence, I assume that the values for the colorbars in Figure 7 should actually say ≤ -0.06 and ≥ 0.06.

Lines 213-214: Wouldn't this oppose the claim that P11 is a typical example for the whole dataset?
Lines 236-237: I find this concept of dry and wet low-water periods confusing. What is meant by a wet low-water period? Is it a comparably high GWL during the time period (JAS) or does the term wet refer to high precipitation? I think further clarification and maybe a different term could help. Maybe something like "most severe / least severe low-water period" instead of "driest / wettest low-water period".

Figure 8: According to the lines previously referred to, the titles for d1 and e1 should be (if you stick to this term) "three driest low-water periods" and "three wettest low-water-periods".

Line 246: Wouldn't that be expected, because evapotranspiration depends on both temperature and water availability?

Conclusions – last paragraph: Another limitation is that all inputs other than P and T are neglected. I think this is also an advantage of the study, since it puts a clear focus on these two inputs, but it should still be mentioned, since the hydrologic cycle is very complex and a lot more things influence water tables other than P and T.

---

## Author Comment (AC2)

We thank the reviewer Bastian Waldowski for his thorough review of our paper. We are happy to read that it is concise, meaningful and well structured. We comment on all relevant points and suggestions of the reviewer in the following and want to thank the reviewer for this very detailed and constructive comments, which help to clarify some important details. In large parts we directly follow the suggestions and adapted our manuscript accordingly. The original review text is in black color, our own comments and answers are colored in blue. Black Line numbers refer to the initially submitted version, blue line numbers to the revised version, which will be submitted after public discussion.

Kind regards on behalf of all authors

Andreas Wunsch

**Review of: Towards understanding the influence of seasons on low groundwater periods based on explainable machine learning**

The paper gives interesting insights into seasonal patterns of groundwater levels, with the focus on low groundwater level time periods. I think the paper is well written and I have read it with interest. The authors use a machine learning method (LRP) that helps them to evaluate which impact the two meteorological inputs they use (precipitation and temperature) have on low groundwater periods within Germany. They have basically found that average temperature has a bigger impact on low groundwater levels than average precipitation, but single precipitation events show the strongest impact. They have found memory effects, like strong precipitation during the preceding winter, to be of less importance.

I think the motivation for this work is very clear, as we obviously want to deepen our understanding about low groundwater periods and the relevant inputs and timeframes behind them. The paper is well structured and concise and the figures are carefully designed and meaningful. The choice which figures are included in the paper and which are shared as supplementary material was made well and the supplementary material is useful and clearly structured. The title is appropriate and the language is good. The LRP method is presented in a plausible way, but I cannot give an in-depth evaluation on that, since I do not come from a machine learning research background. I think that the paper is a good contribution to HESS and recommend that it will be published after minor revisions, which I will elaborate on in the following. I will start with some general comments which include my most essential remarks and then I will get into detailed comments where I reference specific lines.

General comments:

One key conclusion of this work is that temperature is on average more important than precipitation (stated in the abstract and lines 185 and 269). I do not fully understand where this conclusion comes from. I assume one could get to that conclusion from Figure 5, because values are more distributed along the y-axis than the x-axis? I do not think this is clearly stated within the paper (at least not in a way that reached me) and the sentence in line 185 "LRP identifies T as on average more important than P (thus T is on top of the plot)" confuses me. How does it identify T as more important on average? Could this maybe be expressed / quantified by a single value that is then given to the reader?

This is basically concluded from the analysis shown in Figure 6 (but relates also to the analysis in figure 5 as mentioned in the text). The importance on average is derived from the mean absolute value of all relevance scores. We did not explain this properly and added a clarifying statement (L 192). Thank you for pointing this out.

I think that the general claim that dry summers have a large impact on low-water periods and wet winters contribute less to these periods is supported by their findings. However, I think this should be relativized, since I would expect that the amount of impact of the preceding winter depends a lot on the catchment and specific location that is observed. Properties of the soil (including the vadose zone) and delayed release of water by other storages like snow could matter much more at different locations

and for different water table depths (majority of the observation wells are at very shallow locations).

This is correct, thank you. We added respective statements in the conclusion and abstract to clarify that we can only conclude relationships with regard to the investigated locations. We, however, did not change respective statements in other parts of the paper, since it should be clear that we discuss the locations investigated in the study. See L. 13, L. 272, L 277.

The term "importance value" is used sometimes within this work, but it is not explained. If it is the same as relevance score, I suggest sticking to one term. If not, I think it needs to be explained.

This is correct, we now stick to the term "relevance score" throughout the manuscript. Thank you.

Detailed comments:

Abstract: I think it should be stated that you use observations wells distributed over Germany and that they are mostly located in very shallow aquifers.

Done. L10

Lines 9-10: I think it should be stated that you use precipitation and temperature as input data.

Done. L10

Line 15: Writing "more important than precipitation" instead of "the more important variable" would improve understandability.

Done.

Lines 26-28: I disagree with this sentence. Groundwater levels play an important role when estimating recharge, but they do not represent recharge.

Actually, we agree, and we think this is exactly what is stated here already. Nevertheless, we elaborated this section a little bit and hope it is clearer now. This answer includes also the two following suggestions. See LL. 24-36

Lines 28-29: I suggest "recharge can be approximated by" instead of "recharge is".

Compare answer above.

Lines 29-30: I suggest "changes in soil moisture and groundwater storage".

Compare answer above.

Figure 1 - legend: Without the information in the brackets (n = 18) etc. and with the unit directly behind the number (e.g., < 3 m) I would have understood this legend quicker.

Thank you, we changed the way the unit is shown as suggested, however, the information in brackets in still provided, because of its importance and in absence of a better place to present this information. We hope this improves readability nevertheless.

Figure 3 – caption: I suggest removing brackets around "at all 24 locations".

The caption is now adapted to be clearer, thank you.

Lines 165- 168: I agree and would like to add that higher temperatures in winter can also cause the snow storage to melt, which might also play a role with respect to the positive contribution of T.

Thank you, we added this aspect to our manuscript. L. 172

Lines 172-174: If the model learns that conditions one year earlier are important, why does it learn that for fall, but not for winter?

Good question and, unfortunately, we cannot state any reason for this or derive a causality. As elaborated in the manuscript, it is surprising that the connection to the preceding winter is so small compared to the other seasons. At the same time the importance of the preceding fall might indicate that the model can capture some accumulating effects, however, this is not possible for multiple years, because of the approach we chose (limited input time window).

Lines 177-179: I needed to reread this sentence many times, because of the second part of it. I think I understood it now and would recommend writing "it cannot be seen that higher T during dry periods lead to predominantly lower groundwater levels". The way it was phrased before it did not make sense to me, because you always refer to water levels during the low-water period (JAS) in Figure 5

(at least that's how I understood it).

Thank you for pointing out. Unfortunately, your suggestion does not exactly express what was intended from our side. To clarify: we analyze P and T in the four seasons, with respect to their influence in dry periods (Jul-Aug-Sep). Therefore, we now state: "[…] it cannot be seen that higher **summer** T lead to predominantly lower groundwater levels in dry periods." We hope this clarifies the sentence. Compare L. 185

Figure 5: I think it should be stated what we are looking at here (relevance score).

Done, thank you.

Figures 5 and 6: I think it should be clarified what exactly one dot represents (e.g. one dot per week and location). I think there should be actual values at the colorbar.

Thank you for this suggestion. Unfortunately, showing absolute values at the colorbar is not possible, because the scale varies strongly between the locations. However, we now explain what one dot represents in the respective figure captions. Thanks for pointing out.

Line 211: Judging from this sentence, I assume that the values for the colorbars in Figure 7 should actually say ≤ -0.06 and ≥ 0.06.

Actually, the colorbars end with an arrow-shape, meaning that all values above/below have the respective color. The indicated value is correct.

Lines 213-214: Wouldn't this oppose the claim that P11 is a typical example for the whole dataset?

No, because every location has some unique features (this is one of P11). However, we softened our claim in L. 208. We hope this is clearer now.

Lines 236-237: I find this concept of dry and wet low-water periods confusing. What is meant by a wet low-water period? Is it a comparably high GWL during the time period (JAS) or does the term wet refer to high precipitation? I think further clarification and maybe a different term could help. Maybe something like "most severe / least severe low-water period" instead of "driest / wettest low-water period".

Thank you for pointing out. You are correct, the terms are indeed a bit unwieldy. We therefore added your suggested explanation to the text in L. 244, and hope that is clear now. However, we still stick to the term wet and dry. In our opinion both options (yours and ours) would be valid.

Figure 8: According to the lines previously referred to, the titles for d1 and e1 should be (if you stick

to this term) "three driest low-water periods" and "three wettest low-water-periods".

Done. We hope it is clearer now.

Line 246: Wouldn't that be expected, because evapotranspiration depends on both temperature and

water availability?

Thank you for pointing out. This is correct and we added this to our text in L. 255.

Conclusions – last paragraph: Another limitation is that all inputs other than P and T are neglected. I think this is also an advantage of the study, since it puts a clear focus on these two inputs, but it should still be mentioned, since the hydrologic cycle is very complex and a lot more things influence water tables other than P and T.

You are correct. We added an according statement to the conclusions. Thank you for mentioning. Compare LL 282ff.